# Traditional Chinese Medicine Rhodiola Sachalinensis Borissova from Baekdu Mountain (RsB^BM^) for Rheumatoid Arthritis: Therapeutic Effect and Underlying Molecular Mechanisms

**DOI:** 10.3390/molecules27186058

**Published:** 2022-09-16

**Authors:** Yinghui Ma, Jinbei Zhang, Huan Yu, Yanfei Zhang, Huifeng Zhang, Chengyi Hao, Lili Zuo, Nianqiu Shi, Wenliang Li

**Affiliations:** 1College of Pharmacy, Jilin Medical University, Jilin 132106, China; 2College of Public Health, Jilin Medical University, Jilin 132013, China; 3College of Pharmaceutical Science, Yanbian University, Yanji 133002, China; 4Jilin Collaborative Innovation Center for Antibody Engineering, Jilin Medical University, Jilin 132013, China

**Keywords:** rheumatoid arthritis (RA), traditional Chinese medicines (TCMs), natural products, anti-inflammatory activities, NF-κB and RANK/RANKL/OPG signaling pathways

## Abstract

The lack of effective rheumatoid arthritis (RA) therapies is a persistent challenge worldwide, prompting researchers to urgently evaluate traditional Chinese medicines (TCMs) as potential clinical RA treatments. The present investigation was conducted to evaluate the therapeutic effects and potential molecular mechanisms of the active components isolated from TCM Rhodiola sachalinensis Borissova from Baekdu Mountain (RsB^BM^) using an experimental adjuvant arthritis model induced by injection of rats with Freund’s complete adjuvant. After induction of the adjuvant arthritis rat model, the extract-treated and untreated groups of arthritic rats were evaluated for RsB^BM^ therapeutic effects based on comparisons of ankle circumferences and ELISA-determined blood serum inflammatory factor levels (TNF-α, IL-1β, and PGE2). In addition, the joint health of rats was evaluated via microscopic examination of hematoxylin-eosin-stained synovial tissues. Furthermore, to explore whether NF-κB and RANK/RANKL/OPG signaling pathways participated in observed therapeutic effects from a molecular mechanistic viewpoint, mRNA and protein levels related to the expression of nuclear factor kappa-B (NF-κB), osteoprotegerin (OPG), and receptor activator of nuclear factor kappa-Β ligand (RANKL) were analyzed via quantitative RT-PCR and Western blot analysis, respectively. Treatment of arthritic rats with the extract of RsB^BM^ was shown to reduce ankle swelling, reduce blood serum levels of inflammatory factors, and alleviate arthritis-associated synovial inflammation and joint damage. Moreover, an RsB^BM^ 50% ethanol extract treatment inhibited bone destruction by up-regulating OPG-related mRNA and protein expression and down-regulating RANKL-related mRNA and protein expression, while also reducing inflammation by the down-regulating of the NF-κB pathway activity. The results clearly demonstrated that the extract of RsB^BM^ alleviated adjuvant arthritis-associated joint damage by altering activities of inflammation-associated NF-κB and the RANK/RANKL/OPG signaling pathways. Due to its beneficial effects for alleviating adjuvant arthritis, this RsB^BM^ 50% ethanol extract should be further evaluated as a promising new therapeutic TCM treatment for RA.

## 1. Introduction

Rheumatoid arthritis (RA) is a chronic and systemic autoimmune disease with the main clinical manifestations of erosive and symmetrical polyarthritis pathological chronic inflammation of the synovial membrane and pannus that gradually destroys articular cartilage and bone, eventually leading to joint deformity and loss of function [1]. Systemic symptoms of RA include fever, fatigue, loss of appetite, weight loss, night sweats, general malaise, etc., while typical joint-related symptoms include joint pain, swelling, morning stiffness, and deformity. Diagnosis of the disease can be challenging, due to the co-occurrence of atypical disease manifestations, such as pleurisy and valvitis, as well as a lack of clear understanding by clinicians of extremely complex processes involving hereditary, hormonal, and environmental factors that underlie RA initiation and progression. For example, the disease is more likely to occur in people who have been infected with certain viruses, people with abnormal sex hormone levels, smokers, and those with immediate family members with histories of RA [2,3,4].

Comprehensive RA therapeutic goals emphasize early diagnosis, early treatment, standardized treatment, and close monitoring in order to achieve clinical remission, maintain low clinical activity, delay disease progression, reduce disability severity, and preserve the quality of life. Nevertheless, RA is an incurable and chronically remitting, progressive disease that is difficult to manage with current therapies over the course of a patient’s lifetime. At present, several categories of drugs are used most often to treat RA [5]. The first category, non-steroidal anti-inflammatory drugs (NSAIDS) that include indomethacin, ibuprofen, piroxicam, celecoxib, nabumetone, nimesulide, etc., have antipyretic, analgesic, and anti-inflammatory effects. For patients with active RA, NSAIDS can alleviate symptoms, reduce signs of inflammation, and improve joint function, but these drugs cannot prevent new bouts of inflammation. Common adverse reactions include nausea, vomiting, epigastric pain, peptic ulcer bleeding, renal damage, rash, decreased blood cell counts, etc. The second category, glucocorticoids, can quickly relieve clinical symptoms, but their long-term use can lead to metabolic disorders affecting the metabolism of water, sodium, sugar, fat, and protein and can increase patient risks of contracting serious infections, osteoporosis, cataracts, and other adverse reactions. Commonly used drugs include triamcinolone acetonide, betamethasone, etc. The third category, disease-modifying antirheumatic drugs (DMARDs), form the cornerstone of RA treatment since these drugs are used worldwide as first-line RA treatments such that DMARDs are typically administered as soon as possible after the patient is diagnosed with RA. Commonly used drugs include methotrexate, leflunomide, sulfasalazine, isilamod, hydroxychloroquine, and so on. In the fourth category, biological agents such as TNF antagonists, can alleviate joint inflammation symptoms and prevent joint destruction. These agents mainly include infliximab, etanercept, and adalimumab. The main adverse reactions associated with these drugs include infections with pathogenic bacteria (e.g., *Mycobacterium tuberculosis*), fungal infections, and opportunistic infections. In addition, other biological agents, such as IL-6 receptor antagonists (e.g., tocilizumab), may be used when other RA treatments are ineffective or poorly tolerated. Common adverse reactions include decreased blood cell count, increased blood cholesterol level, and increased susceptibility to infections. For RA patients who have not yet been treated with traditional chemical synthetic joint drugs, it is recommended that the patients receive this type of drug either in combination with biologic agents (e.g., etanercept, infliximab, adalimumab, tocilizumab, etc.) or in combination with targeted synthetic biologic agents (e.g., tofacitinib, baricitinib, etc.) [6,7,8,9].

RA treatment most frequently involves the administration of salicylate drugs or immunosuppressive agents that can trigger side effects during long-term use [10,11]. As a traditional Chinese cultural treasure, traditional Chinese medicine (TCM) has unique advantages of long-lasting therapeutic effects and few adverse effects. In fact, TCMs that are used widely for RA treatment include Tripterygium wilfordii Hook. f., Heracleum hemsleyanum Diels, Aconitum carmichaelii, etc. to alleviate symptoms of joint swelling and pain, reduce inflammation, and slow joint destruction [12,13]. Another TCM used for RA treatment, Rhodiola sachalinensis Boriss. is a perennial herb belonging to the genus Rhodiola of the Crassulaceae family. More than 70 subvarieties of RsB grow in China and account for 81% of RsB subvarieties found worldwide. Among them, Rhodiola sachalinensis Borissova (Boriss.) from the Baekdu Mountains region (RsB^BM^) is popular in traditional medical systems in China to treat inflammatory diseases, relieve joint pain and promote the body’s immune function [14]. Due to unique and extreme environmental and climatic conditions on the plateau where RsB^BM^ grows, these plants may have evolved to synthesize distinctive active components that help them survive in that region, including flavonoid compounds. In fact, results of a previously reported in vitro study demonstrated that after flavonoids from RsB^BM^ were extracted and purified, they were found to possess anti-inflammatory, antioxidant, and immunoregulatory activities [15,16]. Based on those results, here we determined whether a 50% ethanol extract of RsB^BM^ that mainly contains flavonoids could alleviate inflammation using a rat adjuvant arthritis model to simulate human RA. This research is valuable since it explores a new potential application for a known TCM RsB^BM^ and also characterizes active TCM components that may provide therapeutic benefits to RA patients.

TCM treatments for RA are limited, due to their complex formulations and unclear mechanisms of action, warranting in-depth investigations to characterize TCM active components and link their therapeutic effects to RA-related signaling pathways. One such pathway, the RANK/RANKL/OPG signaling pathway, plays an important role in bone metabolism, whereby numerous cytokines, growth factors, and hormones interact with RANKL and OPG proteins to regulate bone metabolism. Consequently, this signaling pathway has become a research hotspot in recent years, with three proteins in this pathway found to have key roles in the regulation of bone metabolism, namely RANK, RANKL, and OPG [17,18,19,20]. RANK (receptor activator of nuclear factor kappa-Β ligand), a member of the tumor necrosis factor receptor family, is a transmembrane protein and the only known receptor for RANKL, whereby RANK binds to RANKL to promote osteoclast differentiation. OPG (osteoprotegerin), a secreted glycoprotein also known as osteoprotective protein, can competitively block RANKL binding to RANK to inhibit osteoclast differentiation and thereby inhibit bone resorption [21,22,23]. Nuclear factor κB (NF-κB), a protein belonging to a family of transcription factors, plays an essential role in the pathogenesis of RA and various other inflammatory diseases to stimulate both expressions of numerous inflammatory proteins and the cellular release of excessive amounts of pro-inflammatory cytokines and tissue-destructive enzymes [24,25,26]. Therefore, here we focused on RA-associated NF-κB and the RANK/RANKL/OPG signal pathways in order to explore potential molecular mechanisms of action of the 50% ethanol extract of RsB^BM^ on RA. Ultimately, the results of this work provide a theoretical foundation to guide the future discovery of new applications for TCMs and methods for creating nano-particle-based TCM preparations.

## 2. Results

### 2.1. Effect of the 50% Ethanol Extract of RsBBM on Rat Ankle Joint Swelling

After the rat RA model was established, rats initially exhibited ankle joint swelling, claudication, and loss of appetite. Notably, intragastric administration of this extract to arthritic rat groups led to different degrees of improvement (relative to that observed for untreated arthritic rats) as based on ankle joint circumference measurements taken prior to arthritis induction (0 weeks), 1 week after arthritis induction (day 7 of the extract administration), and 2 weeks after arthritis induction (day 14 of the extract administration) (Figure 1). Analysis of the results revealed that the average values of the ankle joint circumferences across the groups prior to arthritis induction were basically the same. By contrast, after arthritis was induced in rats of all groups (except the control group), ankle joint circumferences were increased relative to the control group as an indicator of joint swelling, thus demonstrating that the arthritis model was created successfully. After the rats in the extract-treated groups received high, medium, and low doses of the 50% ethanol extract of RsB^BM^, the circumferences of the rat ankle joints in all three groups were significantly reduced relative to the ankle joint circumferences of the untreated arthritic rat group (*p* < 0.05), thus indicating that 50% ethanol extract of RsB^BM^ treatments reduced ankle joint swelling in arthritic rats. Moreover, the beneficial effects of the extract increased with increasing treatment time and increasing dose, whereby the beneficial effect was most pronounced in the AA+F (4 mg/mL) group (*p* < 0.01). Nevertheless, administration of high doses of the extract compounds may lead to side effects that could offset beneficial effects, warranting further study.

### 2.2. Effects of the 50% Ethanol Extract of RsB^BM^ on Serum Cytokine Levels

At 24 h after administration of the last dose of the extract, blood was taken from the hearts of rats and then serum levels of inflammatory factors TNF-α, IL-1β, PGE2, and signaling pathway factors RANKL and OPG were determined. Notably, serum levels of inflammatory factors TNF-α, IL-1β, and PGE2 in extract-treated arthritic rat groups were lower than corresponding levels in the untreated arthritic rat group. Thus, these results indicated that the 50% ethanol extract of RsB^BM^ could inhibit the expression of inflammatory factors in arthritic rats, thereby achieving an anti-inflammatory effect. OPG, an osteoprotective protein, is a secreted glycoprotein that can inhibit bone resorption by binding to RANK to competitively inhibit RANKL binding to RANK. However, in untreated arthritic group rats, the ratio of RANKL/OPG was increased relative to ratios of extract-treated arthritic rats, indicating that the 50% ethanol extract of RsB^BM^ treatment inhibited bone resorption to protect bones and joints from arthritis-induced damage (Figure 2).

### 2.3. Effects of the 50% Ethanol Extract of RsBBM on mRNA-Level Expression of Arthritis-Associated Genes

In terms of bone protection, as compared with the untreated arthritic rat (AA) group, extract-treated arthritic rat groups exhibited increased expression of OPG mRNA, decreased expression of RANKL mRNA, and alleviation of adjuvant-induced joint damage. In addition, NF-κB pathway activation, which reflects the known role of this pathway in regulating the expression of inflammatory factors, was reduced in extract-treated arthritic rat groups. These results thus suggest that NF-κB pathway activation plays an important role in inflammation and immune regulation and that the 50% ethanol extract of RsB^BM^ can reduce NF-κB pathway activation to alleviate arthritic joint damage (Figure 3).

### 2.4. Effects of the 50% Ethanol Extract of RsBBM on Expression of Arthritis-Associated Proteins

To further explore the mechanisms underlying the therapeutic effect of the extract on adjuvant-induced arthritis in rats, Western blot analysis was conducted to measure protein expression levels of OPG, RANKL and NF-κB in rat tissue samples (Figure 4 and Figure 5). Levels of OPG protein in specimens of extract-treated arthritic rat groups were significantly greater than that of the untreated arthritic (AA) group, while levels of RANKL protein in extract-treated arthritic rat groups were lower than that of the AA group, indicating that the 50% ethanol extract of RsB^BM^ protected the bones of the arthritic rats. Moreover, levels of NF-κB protein in tissues of flavonoid-treated arthritic rat groups were lower than that of the AA group, indicating that the 50% ethanol extract of RsB^BM^ may reduce NF-κB pathway activation, thereby reducing the strength of the inflammatory response.

### 2.5. Histopathological Examination

Ankle joint structures of normal (control, non-arthritic) rats were complete, with orderly cell distributions, normal nuclei, lack of pathological findings, and complete synovial epidermal structures with smooth surfaces. By contrast, the ankle joint structures of AA group rats revealed severe joint damage manifesting as synovial cell aggregation with structurally abnormal nuclei that were enlarged as compared to normal nuclei. By contrast, stained ankle joint tissues of the three extract treatment groups revealed that the 50% ethanol extract of RsB^BM^ alleviated inflammation-induced synovial damage but did not restore synovial tissues to a normal state (Figure 6). Furthermore, scores for inflammatory infiltration, synovial hyperplasia, cartilage destruction, and bone erosion of extract-treated groups were reduced as compared to the corresponding scores for the AA group, with the most pronounced beneficial effects observed for the group receiving the highest extract dose, the AA+F (4 mg/mL) group (Table 1).

## 3. Discussion

RA is a chronic autoimmune disease. At present, drugs used to treat RA mainly include non-steroidal anti-inflammatory drugs (NSAIDS), corticosteroids, and disease-modifying antirheumatic drugs (DMARDs). However, when patients take these drugs long-term, they often experience side effects involving the gastrointestinal tract and cardiovascular system or other complications (e.g., osteoporosis and high blood pressure). TCM, a cultural treasure of China, has contributed much clinical knowledge that has fostered international pharmaceutical development all over the world. In this study, we evaluated a 50% ethanol extract of Rhodiola sachalinensis Boriss. from Baekdu Mountain (RsB^BM^) for efficacy as a therapeutic RA treatment using a rat model of RA generated by the inoculation of rats with a complete Freund’s adjuvant. Using this model, the therapeutic effects of the extract of this TCM were assessed based on the reduction of toe swelling, reduction of serum levels of inflammatory factors, and repair of joint damage in treated versus untreated arthritic rats.

The complexity of TCM is reflected in the diversity of its chemical components and the interaction of various components. The synergistic effect is the key to improving the efficacy of TCM and reducing toxicity and side effects. Therefore, TCM is not a single chemical component. Due to the complexities of TCM formulations, which are often composed of complex mixtures of chemical components that act via unclear mechanisms, efforts to develop TCMs as treatments for human diseases are limited, resulting in a huge waste of Chinese medicinal resources. Therefore, research efforts to identify mechanisms underlying beneficial TCM effects have intensified in recent years, resulting in numerous studies demonstrating that molecular mechanisms underlying beneficial TCM effects on RA mainly involve PI3K-Akt, cAMP, MAPK, NF-kB, and other signal molecules [27,28,29,30].

Members of the NF-κB protein family can selectively bind to the B cell kappa-light chain enhancer to regulate the expression of numerous genes. In fact, NF-κB protein family members are found in almost all animal cells, due to their roles in cell responses to external stimuli such as cytokines, radiation, heavy metals, viruses, and so on. Importantly, NF-κB plays a key role in cellular inflammation and immune responses whereby dysregulation of NF-κB activities was linked to autoimmune diseases, chronic inflammation, and many cancers. NF-κB receptor activator (RANK) is a tumor necrosis factor receptor and a central activator of NF-κB. Meanwhile, the osteoprotective agent osteoprotegerin (OPG) is a decoy receptor homolog of the RANK ligand (RANKL), which inhibits RANK activity by binding to RANKL to thereby regulate NF-κB pathway activity. Based on these observations, several researchers have worked to determine whether the RANK/RANKL/OPG signaling pathway is associated with TCM therapeutic effects for alleviating RA [31,32,33,34]. In a similar vein, here we focused our attention on NF-κB and the RANK/RANKL/OPG signaling pathways in order to reveal the mechanism of action associated with arthritis-alleviating effects of a 50% ethanol extract of RsB^BM^ using a rat adjuvant-induced arthritis model of human RA. The results revealed that treatment of arthritic rats with the extract led to increased expression of OPG and decreased RANKL expression, while also significantly inhibiting NF-κB pathway activity.

In this study, complete Freund’s adjuvant was used to establish a rat adjuvant arthritis model to simulate human RA. A 50% ethanol extract that is rich in flavonoids such as salidroside of TCM RsB^BM^ was used to treat arthritic rats and then possible mechanisms underlying observed beneficial treatment effects were explored using RT-PCR and Western blot analysis. The results revealed that treatment of arthritic rats with the extract of RsB^BM^ could reduce serum levels of arthritis-associated inflammatory factors by inhibiting the NF-κB pathway. In addition, extract-treated arthritic rats exhibited reduced toe swelling and reduced articular cartilage erosion (as compared to corresponding characteristics of untreated arthritic rats) that were associated with treatment-induced OPG level increases resulting from altered RANK/RANKL/OPG signaling pathway activity. The results of the present work revealed that the RsB^BM^ holds promise as an RA treatment, and developing the medicinal components of TCM into novel nano-drugs to improve their stability and bioavailability, thereby enhancing the therapeutic effect of natural medicines could guide future research and development of TCM-based treatments for human diseases [35,36].

## 4. Materials and Methods

### 4.1. Chinese Herbal Extract

Roots and stems of Rhodiola sachalinensis Boriss. were purchased from Beijing Tongrentang Pharmacy (Jilin Province, China) and identified through morphological examination conducted by Professor Jing-hua Li (Crude Drugs Teaching and Research Section of Jilin Medical University). A voucher specimen (RsB-20190306) was deposited in the herbarium of College of Pharmacy of Jilin Medical University. The active components of RsB^BM^ were obtained by extraction and purification with 50% ethanol. The extraction process was that 50% ethanol was used as the extraction solvent, the solid–liquid ratio was 1:35, and the ultrasonic extraction was carried out at 60 °C and 200 W for 35 min and the purification process was that the ratio of diameter to height of the chromatographic column was 1:16, 2 BV was loaded at a flow rate of 2 BV/h, and 2.5 BV was eluted with 50% ethanol. After extraction and purification, the flavonoid ingredients reached 81.6% in the active components by UV spectrophotometry. Next, we took this Chinese herbal extract as our research object because it was rich in flavonoids, and it is represented by F. These active ingredients have a wide range of pharmacological effects, such as antioxidant, anti-inflammatory, and anti-cancer [37,38,39].

### 4.2. Chemicals and Reagents

Phosphate-buffered saline (PBS) (P1010), bicinchoninic acid (BCA) protein concentration determination kit (PC0020), radioimmunoprecipitation assay (RIPA) lysis solution (P0010), and differentiation solution (G1860) were all purchased from Solarbio (Beijing, China); 5× non-denaturing, non-reducing protein loading buffer (B130043) was supplied by BioNtech Co., Ltd. (Mainz, Germany); sodium dodecyl sulfate-polyacrylamide gel electrophoresis (SDS-PAGE) gel preparation kit (BL508A) was provided by Biosharp Co., Ltd. (Shanghai, China); Tris base (T1378), acetone (179973), paraformaldehyde (P6148), hematoxylin stain (03971), and diethyl pyrocarbonate (DEPC) (SM-V900882) were obtained from Sigma Co., Ltd. (St. Louis, MO, USA); glycine (NC-S0002-500) and sodium lauryl sulfate (NC-0227-500) were purchased from Nachuan Biotechnology Co., Ltd. (Harbin, China); Tween 20 (1247ML100) was purchased from BIOFROXX (Einhausen, Germany); Fekete Ultra-sensitive ECL luminescent fluid (MA0186-L) was purchased from Meilunbio Co., Ltd. (Dalian, China); Western primary antibody and secondary antibody removal fluid (P00258) were supplied by Beyotime Co., Ltd. (Shanghai, China); xylene (A509900), eosin (E607321), neutral gum (E675007), and ethylenediaminetetraacetic acid (EDTA) decalcification solution were provided by Shenggong Biological Engineering Co., Ltd. (Shanghai, China); OPG (AB73400), RANKL (AB239607) and NF-κB (AB16502) were obtained from Abcam Co., Ltd. (Cambridge, UK); OPG (ml525574), RANKL (ml003065), PGE2 (ml003036), IL-1β (ml037361), and TNF-α (ml002859) ELISA kits were purchased from Mlbio Co., Ltd. (Shanghai, China).

### 4.3. Therapeutic Administrative Protocol

The male Sprague Dawley (SD) rats were purchased from the Animal Experimental Center of Harbin Medical University (Harbin, China). Fifty rats (mass/rat of 200–230 g) were randomly selected and raised under suitable temperature and humidity conditions. After rats were allowed to adapt to the living environment, they were randomly assigned to 5 groups, including control, adjuvant arthritis (AA), adjuvant arthritis (AA)+F (2 mg/mL), adjuvant arthritis (AA)+F (4 mg/mL), and adjuvant arthritis (AA)+F (10 mg/mL) groups. Ankle joint circumference of the injected one was measured once prior to injections of rats with adjuvant to generate the arthritis model. Except for the control group, the other four groups of rats were injected intracutaneously in the left hind toe with 0.1 mL of complete Freund’s adjuvant to induce inflammation. After 14 days, ankle joint circumferences of rats in each group were measured again and then each rat in AA+F (2 mg/mL) group, AA+F (4 mg/mL) group, and AA+F (10 mg/mL) groups was orally administered 2 mL of flavonoids once per day for 14 days. Twenty-four hours after rats received the final flavonoid dose, ankle joint circumference was measured, blood was taken from the heart, and both ankle joints were harvested from each rat.

### 4.4. Serum Cytokines Measurements

After completion of RsB^BM^ extract administration, blood was collected and then serum cytokine levels were measured via ELISA. To obtain serum, rat blood samples were centrifuged (SORVALL ST8/8R type, Thermo, Waltham, MA, USA) at 3000 rpm for 20 min at 4 °C and then the supernatant was retained. To generate standard curves for use in ELISAs, absorbances of 50 μL volumes of different concentrations of rat OPG, RANKL, PGE2, IL-1β, and TNF-α standards were determined at 450 nm using a microplate reader (MULTISKAN FC, Thermo, Waltham, MA, USA). Next, rat serum samples were assayed via ELISA and then absorbance values were read using the same method as was used for ELISA standards mentioned above.

### 4.5. RNA Extraction, cDNA Reverse Transcription, and Quantitavive Real-Time PCR

After completion of RsB^BM^ extract administration, total RNA was prepared from blood and ankle joint tissues obtained at that time using TRIzol reagent. Next, RNA was reverse-transcribed using an American SBP SelectCycler II Gradient Gene Amplifier (SBT9610-230V) according to the manufacturer’s instructions. Thereafter, cDNA was used as template for quantitative RT-qPCR that was performed using a Bio-Rad Real-Time PCR System (CFX96, Bio-Rad Co., Ltd., Hercules, CA, USA). All primers were designed using primer software (Premier 5.0, Toronto, ON, Canada). Primer sequences are listed in Table 2.

### 4.6. Western Blot-Based Detection of Protein Expression Levels

After completion of RsB^BM^ extract administration, synovial tissues (about 20 mg/specimen) were harvested from animals. Next, the tissues were minced using an Automatic Sample Rapid Tissue Grinder (Tissue lyser-24 type, Shanghai Jingxin Technology Co., Ltd., Shanghai, China) and then minced tissues were homogenized in lysis buffer. Thereafter, samples were centrifuged at 13,000 rpm for 10 min at 4 °C and then the supernatant was retained and the protein concentrations in the supernatants were measured using a BCA protein assay kit. Proteins were next separated via SDS-polyacrylamide gel electrophoresis (SDS-PAGE) and then proteins were electroblotted onto polyvinylidene difluoride (PVDF) membranes. In addition, equal amounts of internal reference standards for RANKL, OPG, NF-κB, and GAPDH were loaded into gel lanes, electrophoresed and then electroblotted onto a PVDF membrane. Next, membranes were immersed in 5% blocking solution and then placed on a shaker and incubated with shaking at room temperature for 2 h. PVDF membranes containing electroblotted target proteins or internal reference standards were completely immersed in diluted primary antibodies and incubated overnight at 4 °C. After diluting corresponding secondary antibody stock solutions with blocking solution, PVDF membranes were immersed in diluted secondary antibody and incubated at room temperature for 1 h. Finally, membranes were immersed in enhanced chemiluminescent (ECL) solution in the dark and incubated for 2 min to allow color to develop. Membranes were then analyzed using an automatic chemiluminescence image analysis system to quantify signal intensities associated with antibody-bound protein bands.

### 4.7. Histological Study of Rat Joint Tissues

Rat joint tissues were fixed in 4% paraformaldehyde (PFA) and decalcified with decalcifying reagent. Next, tissue sections were soaked in increasing concentrations of ethanol for 5 min each soak and then sections were soaked in xylene for 2.0 s/soak. Next, sections were immersed in paraffin at 45 °C for 45 min and then placed in a mold and allowed to solidify. Next, paraffin-embedded tissues were sliced (5 μm-thick slices) using a Slicer (Germany Leica RM2016 type, Weztlar, Germany), and then slices were affixed to glass slides and slides were processed according to instructions provided with the slicer. Finally, slides were stained, viewed, and imaged under a microscope (Boaosen Biotechnology Co., Ltd., Beijing, China).

### 4.8. Data Analysis

Statistical analysis was performed via unpaired Student’s *t*-test and expressed as the mean ± standard deviation (SD). Each statistical experiment was repeated in triplicate (*n* = 3). Significance was deemed for *p* < 0.01 and *p* < 0.05.

## Figures and Tables

**Figure 1 molecules-27-06058-f001:**
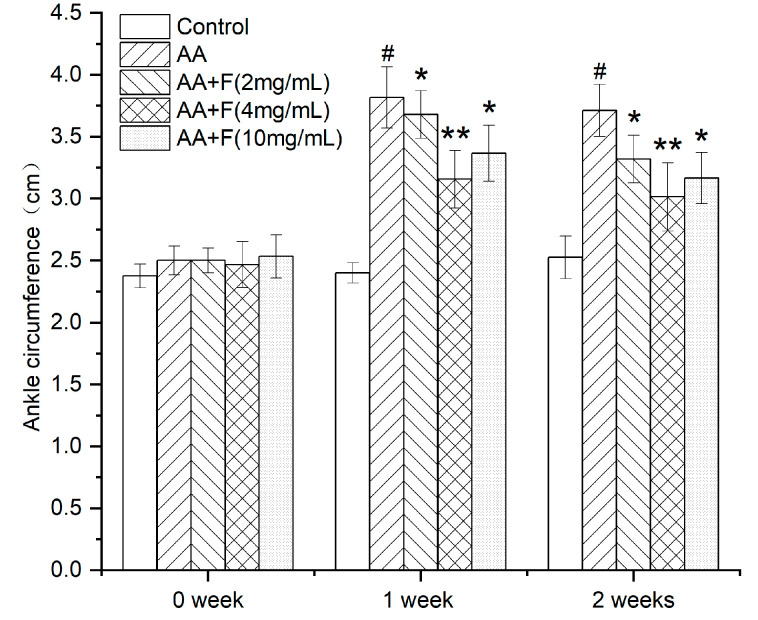
Effects of different treatments on reducing ankle joint circumference of arthritic rats (x ± SD). Error bars represent standard deviations (*n* = 6). As compared with control group, # *p* < 0.05 indicates the arthritis model was established successfully. As compared with the AA group, * *p* < 0.05 indicates significant effects of AA+F (2 mg/mL) and AA+F (10 mg/mL) treatments in alleviating ankle joint swelling in arthritic rats; ** *p* < 0.01 indicates a more significant effect of AA+F (4 mg/mL) in reducing ankle joint circumference of arthritic rats.

**Figure 2 molecules-27-06058-f002:**
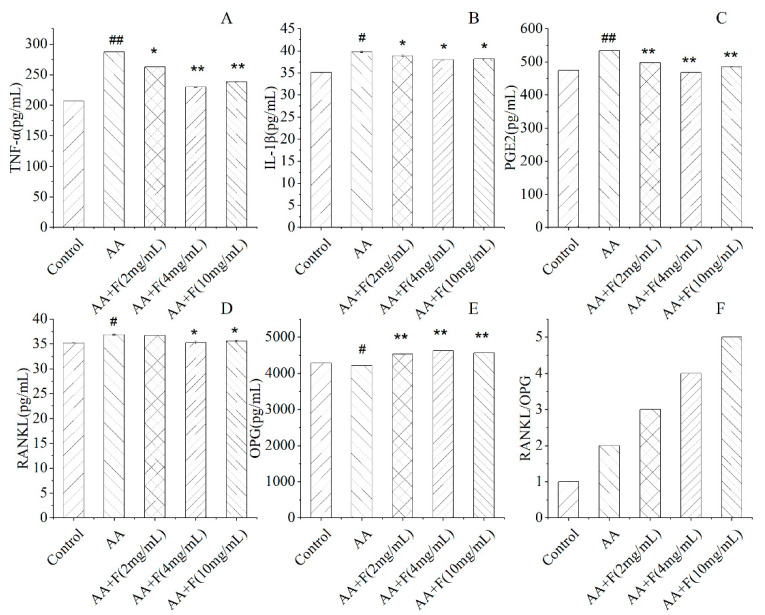
Effects of different treatments on levels of serum cytokines TNF-α (**A**), IL-1β (**B**), PGE2 (**C**), and on levels of signaling pathway proteins RANKL (**D**), OPG (**E**) and RANKL/OPG (**F**) in arthritic rats (x ± SD). Error bars represent standard deviations (*n* = 6). As compared with the control group, # *p* < 0.05 and ## *p* < 0.01 indicate that serum cytokine levels of the untreated arthritis model (AA) group were significantly different from corresponding control group levels. As compared with the untreated arthritis model (AA) group: * *p* < 0.05 and ** *p* < 0.01 indicated significant reduction in serum cytokine levels in the treated arthritis model group.

**Figure 3 molecules-27-06058-f003:**
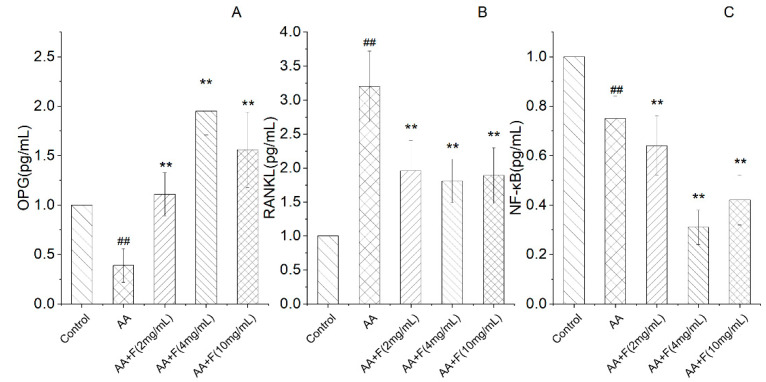
Expression of OPG (**A**), RANKL (**B**), NF-κB (**C**) mRNA as detected by quantitative RT-PCR (x ± SD). Error bars represent standard deviations (*n* = 3). As compared with the control group, ## *p* < 0.05 indicates that mRNA expression levels of OPG and NF-κB decreased and the mRNA expression level of RANKL increased in the arthritis model (AA) group. As compared with the AA group, ** *p* < 0.01 indicates that groups of treated arthritic rats exhibited increased expression of OPG mRNA, decreased expression of RANKL mRNA, and significant inhibition of NF-κB signaling pathway activity.

**Figure 4 molecules-27-06058-f004:**
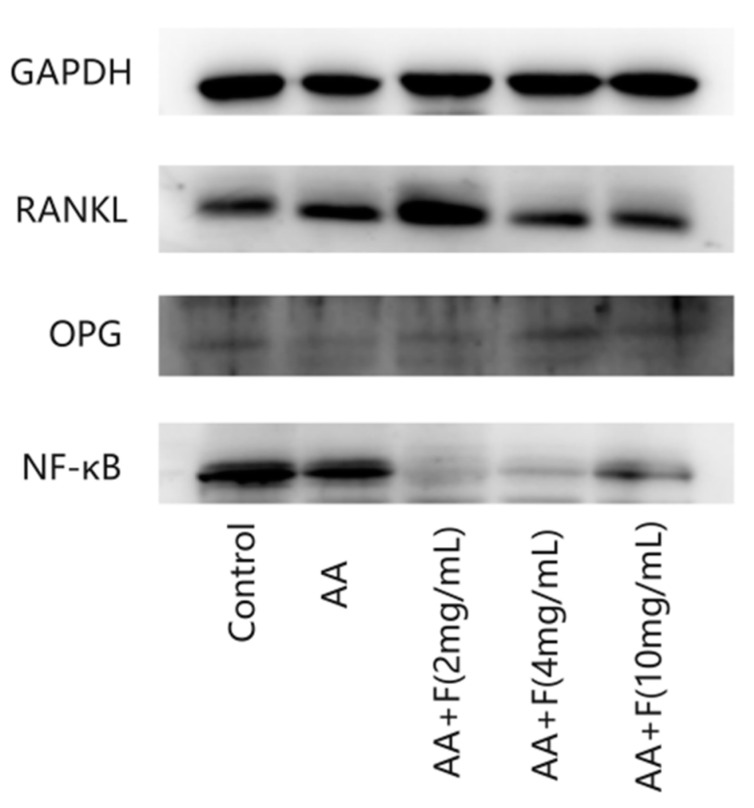
Expression of OPG, RANKL, NF-κB protein detected by Western blot.

**Figure 5 molecules-27-06058-f005:**
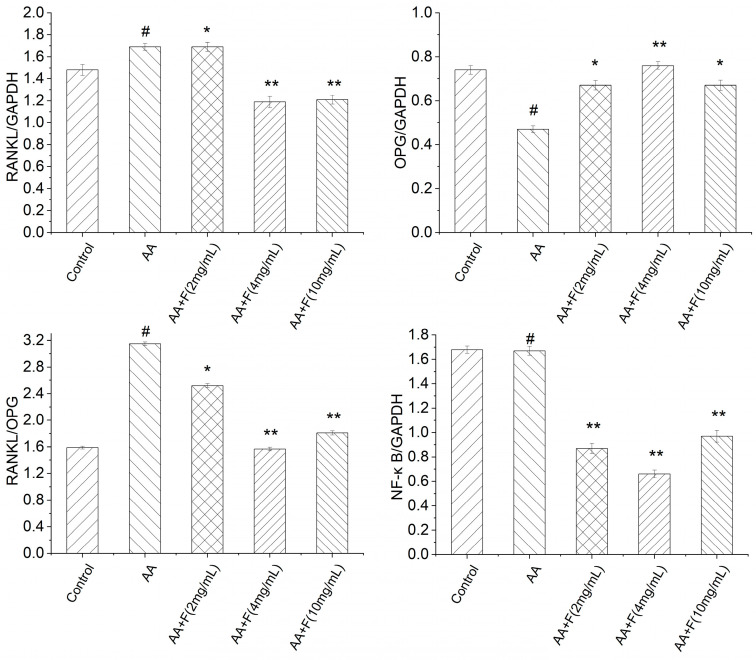
Expression of OPG, RANKL, NF-κB proteins as detected via Western blot analysis (x ± SD). Error bars represent standard deviations (*n* = 3). As compared with the control group, # *p* < 0.05 indicated that the arthritis model (AA) group was established successfully, as evidenced by decreased expression of OPG and NF-κB proteins and increased expression of RANKL protein in the AA group. As compared with the AA group, * *p* < 0.05 and ** *p* < 0.01 indicated that extract-treated arthritis model (AA+F) groups exhibited increased expression of OPG protein and decreased expression of RANKL protein and significant inhibition of the NF-κB pathway.

**Figure 6 molecules-27-06058-f006:**
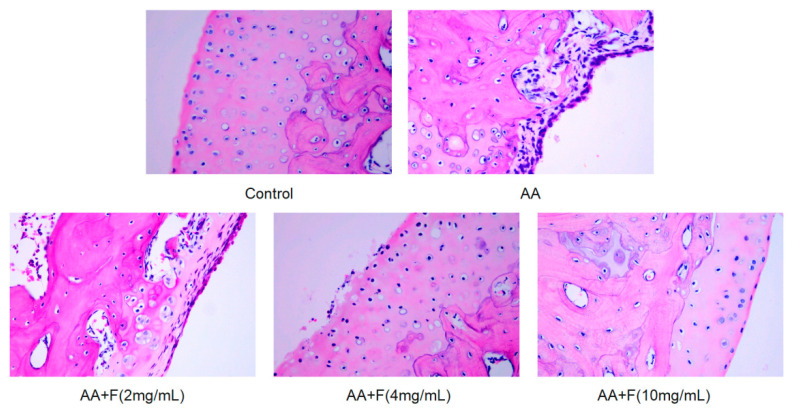
Ankle joint tissues of hind paws of treated and untreated arthritic rats were histologically examined after H&E staining (×400). Joint structures in rats of the AA group were severely damaged, synovial cells were aggregated, and nuclei were structurally abnormal and larger than normal nuclei. Treatment of arthritic rats with the 50% ethanol extract of RsB^BM^ extract led to reduced inflammation-induced synovial damage but did not restore synovial tissues to a normal state.

**Table 1 molecules-27-06058-t001:** Arthritis score sheet (*n* = 6). For each group, scores for inflammatory infiltration, synovial hyperplasia, cartilage destruction, and bone erosion are listed.

	Inflammatory Infiltration	Synovial Hyperplasia	Cartilage Destruction	Bone Erosion	Total Score
Control	0	0	0	0	0
AA	2.8	2.4	2.1	1.8	9.1
AA+F (2 mg/mL)	2.5	1.9	1.0	0.8	6.2
AA+F (4 mg/mL)	1.8	1.4	0.7	1.5	5.4
AA+F (10 mg/mL)	2.7	1.7	0.8	0.7	5.9

**Table 2 molecules-27-06058-t002:** The sequences of the primers.

Gene	Primer Sequence (5′-3′)	Product (bp)
GAPDH-Rat-F	GGTGGACCTCATGGCCTACAT	21
GAPDH-Rat-R	CTCTCTTGCTCTCAGTATCCTTGCT	25
OPG-Rat-F	ACTTGGCCTCCTGCTAATTC	20
OPG-Rat-R	CGCACAGGGTGACATCTATT	20
RANKL-Rat-F	CATCGCTCTGTTCCTGTACTT	21
RANKL-Rat-R	CGAGTCCTGCAAACCTGTAT	20
NF-κB-Rat-F	GGTTACGGGAGATGTGAAGATG	22
NF-κB-Rat-R	GTGGATGATGGCTAAGTGTAGG	22

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
