# Peer review of "Traditional Chinese Medicine Rhodiola Sachalinensis Borissova from Baekdu Mountain (RsBBM) for Rheumatoid Arthritis: Therapeutic Effect and Underlying Molecular Mechanisms"

_molecules, 2022, doi:10.3390/molecules27186058_

Round 1
Reviewer 1 Report (Previous Reviewer 2)
In the respons to reviewer 2, author said as follows; "But in fact the major compounds that I designated as flavonoids was salidroside, which is one kind of flavonoid, not tyrosol and its glycoside, which is a Reviewer 2's mistake here". However, the Salidrosid is the glycoside of tyrosol and its structure does not meet the criterial of flavonoid including flavanol, flavan, flavanone, flavonol, isoflavone, and anthocyanin.
According to IUPAC, "Chemically, flavonoids have the general structure of a 15-carbon skeleton, which consists of two phenyl rings (A and B) and a heterocyclic ring (C, the ring containing the embedded oxygen). This carbon structure can be abbreviated C6-C3-C6. According to the IUPAC nomenclature".

Author Response
Thank you for your professional review comments. I am sorry that the author has a misunderstanding of compound classification. Salidroside and tyrosol should be glycosides and their aglycones. The flavonoids mentioned in the article are actually substances that have been extracted and purified and identified by flavonoid identification and content determination methods, but there are still salidroside that has not been completely removed. Therefore, according to the previous reviewer's opinion, the author changed the extracted and purified product to 50% ethanol extract. I don't know if it is suitable, I hope experts can give guidance and suggestions. If there are any unresolved issues, please contact and advice.
Reviewer 2 Report (New Reviewer)
Professional language revision is mandatory. eg. "Nature products" in Keywords should be Natural products. In the Introduction, RA therapy goals" should read RA therapeutic goals etc
Statements of the adverse effects of NSAIDs, steroids and DMARDs in the Introduction should have appropriate references. You should justify the use of 50 % ethanol and the choice of 2, 4 and 10 mg/ml. Do you have any idea about the LD50 of the extract and the flavonoid component?
How was the the joint circumference measured?
The results appeared not dose-related. The effects produced by 4 mg/kg of F (eg Figure 1) were mostly greater than those of 10 mg/kg, how would you explain this?
Author Response
Thank you for your professional review comments. Now we will answer your questions, and submit the corresponding revisions to the revised draft. If there are any unresolved issues, please contact and advice.
- Content suggested by experts requiring language revision has been revised. Sorry for such mistakes.
- The appropriate references about the statements of the adverse effects of NSAIDs,steroids and DMARDs in the Introduction are listed in references [6-9].
- The use of 50 % ethanol is because that the content of flavonoids in different volume fractions of ethanol was compared by single factor in the previous extraction and purification experiment, it was determined that the content of flavonoids in 50% ethanol was the highest.
- It is reported that the daily dose of Rhodiola sachalinensis Borissova is about 6-12g. According to the conversion of human and rat doses, the content of flavonoids in Rhodiola sachalinensis Borissova, and the pre-experiment results of LD50 of flavonoid extracts, three concentration gradients of 2/4/10 were selected for animal experiments.
- Rhodiola has the functions of tonifying qi and clearing the lungs, nourishing the mind and nourishing the mind, removing astringents to stop bleeding, dispersing blood stasis and reducing swelling. The toxic and side effects are rare. Before the animal test, an acute toxicity test was carried out. Combined with the conversion of human drug dosage, 20mg/kg irrigation was selected. After gastric administration, no obvious abnormality was found in the functions of various organs.
- Mark the ankle joint of each rat, and measure the circumference of the ankle joint with a soft ruler at the designated time after modeling and administration.
- Yes, the results appeared not dose-related, which is different from what I imagined. It may be due to the effect of high-dose administration on the gastrointestinal tract of rats, because it was found in the experiment that individual rats in the high-dose group had occasional diarrhea, and their mental state and activity were not as good as those in the low-dose and medium-dose groups, which may have had influences on therapeutic effect.
Reviewer 3 Report (New Reviewer)
The paper is very interesting since it proposes the study of the effect of an extract of Rhodiola sachalinensis used in traditional Chinese medicine in a rheumatoid arthritis in vivo model. I think it would be convenient to delve into some aspects. The authors should show details of the purification and characterization of the extract , quantification of the main metabolites or marker metabolites of the chemical quality of the extract for example salidroside
The following paragraph should be explained in more detail:
A) And the purification process was that the ratio of diameter to height of the chromatographic col-umn was 1:16, 2 BV was loaded at a flow rate of 2 BV/h, and 2.5 BV was eluted with 50% ethanol.
B) After extraction and purification, the flavonoids ingredients reaches 81.6% in the active components by UV spectrophotometry
C) The chemical components were identified by liquid chromatography, and the most abundant content of salidroside, which is the main active ingredient in RsBBM.
I It is also necessary to develop the discussion. Incorporate citations on previous studies of bioactivities of compounds such as salidroside
Author Response
Thank you for your professional review comments. Now we will answer your questions, and submit the corresponding revisions to the revised draft. If there are any unresolved issues, please contact and advice.
- In the preliminary work of this experiment, we used the ultrasonic extraction method for extraction, and obtained the optimal extraction conditions through the single-factor variable method. AB-8 macroporous adsorption resin was used for purification, and the concentration of the drug solution, the amount of sample loaded, and the concentration of the eluent were investigated, and elution volume and other factors to determine the best purification process. The results showed that the optimal extraction process of the flavonoids was as follows: 50% ethanol was used as the extraction solvent, the solid-liquid ratio was 1:35, and the ultrasonic extraction was carried out at 60 ℃ and 200 W for 35 min. And the optimal purification process was as follows: the ratio of diameter to height of the chromatographic column was 1:16, the concentration of flavonoids in the liquid was 0.05 mg/mL, 2 BV was loaded at a flow rate of 2 BV/h, and 2.5 BV was eluted with 50% ethanol. After purification, the flavonoids ingredients reaches 81.6% determined by UV method. The chemical components contained in the extracted and purified flavonoids were identified by liquid chromatography. Chromatographic separation was achieved on an Agilent Zorbax Eclipse Plus C 18 column (250 mm×6 mm,5 μm) with Methanol-acetonitrile-0.06% phosphoric acid (10:10:80) as the mobile phase. Detective wavelength was set at 275 nm,and the flow rate was 1.0 mL·min-1. And the other peaks include flavonoids such as quercetin, kaempferol and salidroside and its aglycone tyrosol.
Because this part of the experimental process is cumbersome and has been published separately in the Chinese core journals, in order to avoid the suspicion of multiple submissions for one manuscript and the length of the article, it is not explained in detail in this manuscript. I am not sure if it would be more appropriate to provide supplementary material, looking for expert guidance.
- A brief discussion of the bioactivity studies of the main flavonoids is added in the text, and references are attached [27-30]

Round 2
Reviewer 1 Report (Previous Reviewer 2)
The manuscript has been a lot improved. it can be Accepted in present form
This manuscript is a resubmission of an earlier submission. The following is a list of the peer review reports and author responses from that submission.
Round 1
Reviewer 1 Report
The method of extraction and isolation of the plant compounds is not presented. Isolated compounds, including flavonoids, are not detailed in the results. It is spoken in a general way of "flavonoid extract"; is it a raw extract? in the extracts and depending on the solvents used, the compounds obtained vary. In this sense, many other compounds have been tested against AR and have been effective (Diterpenes, Alkaloids, Phenolic Compounds, Triterpenoids, among others), some of which are very common in plants, so how did you ensure that there are only flavonoids in the extract? It is important to characterize the extract. What flavonoids are found in it? On the other hand, the plant was purchased, which prevents the replicability of the experiments, since it is known that some metabolites respond to environmental variation, without forgetting that it is a plant with problems due to the reduction of its habitat and that a solution for its use would be cultivation, but looking for quality control, under what conditions? Answering these questions would substantially modify the article.
Author Response
Thanks to the experts for your valuable comments, the article has been revised and the corresponding questions have been answered, please see the attachment. If there are any unfinished matters, please contact me.

Reviewer 2 Report
The author investigated therapeutic effect of RsBBM on rheumatoid arthritis and underlying molecular mechanisms.
1. In " Among them, Rhodiola sachalinensis Borissova (Boriss.) from the Baekdu Mountains region (RsBBM) is popular in traditional medical systems in China to treat inflammatory disease, relieve joint pain and promote body's immune function [14-15]. ", reference 14 is a research about enhancing iNOS synthesis by hodiola sachalinensis Borissova extracts which indicates enhancing inflammation or immune boosting. The reviewer thinks reference 14 is not appropriate and unrelated reference with the contents and also author mentioned why the current study and reference show a distinct trend.
2. How to prepare the flavonoid extract of RsBBM ? The author should explain the procedure of producing flavonoid extract of RsBBM .
3. Also, the manuscript is lack of the content of the flavonoid extract of RsBBM . Author should analyze total flavonoid amounts and major flavonoid of sample. If could the author should clarify why the author designated the sample as flavonoid extracts including potential bioactive compounds among flavonoids as well as
Author Response

(The authors gave the same response as above.)

Round 2
Reviewer 1 Report
The article can be published.
Reviewer 2 Report
Although the authors explained extraction, purification, and identification methods in the manuscript, the manuscript has a critical flaw. The major compounds the author designated as flavonoids were tyrosol and its glycoside. However, the tyrosol is not a series of flavonoid. It is a phenolic acid. Therefore, the author should thorououtly inspect the manuscript .